# Improvements in Systemic Therapies for Advanced Malignant Mesothelioma

**DOI:** 10.3390/ijms241310415

**Published:** 2023-06-21

**Authors:** Chiara Deiana, Francesca Fabbri, Simona Tavolari, Andrea Palloni, Giovanni Brandi

**Affiliations:** 1Medical Oncology, IRCCS Azienda Ospedaliera, Universitaria di Bologna, 40138 Bologna, Italy; chiarad.kw@gmail.com (C.D.); francescafabbri92@gmail.com (F.F.); andrea.palloni@aosp.bo.it (A.P.); giovanni.brandi@unibo.it (G.B.); 2Department of Medical and Surgical Sciences, University of Bologna, 40138 Bologna, Italy

**Keywords:** mesothelioma, advanced mesothelioma, systemic therapies, novel approaches, immunotherapy, target therapy, TKI, vaccines, antiangiogenic

## Abstract

Malignant pleural mesothelioma (MPM) is a rare and aggressive malignancy associated with poor prognosis and a 5-year survival rate of 12%. Many drugs have been tested over the years with conflicting results. The aim of this review is to provide an overview of current therapies in MPM and how to best interpret the data available on these drugs. Furthermore, we focused on promising treatments under investigation, such as immunotherapy with targets different from anti-PD-1/PD-L1 inhibitors, vaccines, target therapies, and metabolism-based strategies.

## 1. Introduction

Malignant pleural mesothelioma (MPM) is a rare and aggressive malignancy strongly associated with asbestos exposure. Despite the introduction of restrictions on asbestos use starting from the 1970s, the rate of mesothelioma cases in the United States kept increasing until the early 1990s due to the long latency between asbestos exposure and the onset of the disease; in the last years, the incidence has been slowly declining, with 2875 cases reported in 2018 with an incidence rate of 0.7/100,000 people [1,2]. Mesotheliomas are much more common in older people with a median age at diagnosis of 72 years and a poor prognosis with a 5-year survival rate of 12% [3]. Thus, more effective treatments are urgently needed.

This paper aims to offer an overview of current therapies in MPM and of future promising treatments under investigation.

## 2. First-Line Treatments

### 2.1. Chemotherapy

In 2003, the EMPHACIS [4] phase 3 trial established cisplatin and pemetrexed as the standard first-line regimen for unresectable mesothelioma. It proved a significant gain in median overall survival (mOS) for the doublet when compared to cisplatin alone (12.1 vs. 9.3 months, *p* = 0.020), as well as better median progression-free survival (mPFS, 5.7 vs. 3.9 months, *p* = 0.001) and overall response rate (ORR, 41.3% vs. 16.7%, *p* < 0.0001).

A similar phase 3 trial, investigating the combination of cisplatin and a different antifolate, raltitrexed, again proved the superiority of a chemotherapy doublet compared to cisplatin monotherapy (mOS: 11.4 months vs. 8.8 months) [5]. It should be noted, however, that raltitrexed is not registered in many European countries for this indication. 

In elderly or unfit populations, carboplatin is a reasonable alternative to cisplatin to decrease toxicity. There are no randomized trials comparing cisplatin vs. carboplatin in MPM, however, data on systematic review and Expanded Access Program (EAP) show that the combination of carboplatin/pemetrexed is not significantly different from cisplatin/pemetrexed in PFS and OS [6,7]. 

Several attempts have been made to improve the results of the EMPHACIS trial with the use of maintenance therapies, all unsuccessful. A phase 2 trial of Cancer and Leukemia Group B (CALGB) showed that pemetrexed continuation after 4–6 cycles of doublet chemotherapy induction did not improve PFS compared with observation [8], and a randomized phase 2 trial (NVALT19) investigating a switch to maintenance gemcitabine after first-line chemotherapy failed to prove an OS benefit, despite a longer PFS [9]. Since then, several new targets and strategies have been investigated.

### 2.2. Immunotherapy

The combination of anti-programmed cell death protein 1 (PD-1) antibody Nivolumab and anti-cytotoxic T-lymphocyte protein 4 (CTLA-4) Ipilimumab has shown clinical benefit in different tumor types, including mesothelioma.

The randomized phase 3 trial CheckMate 743 [10] investigated this combination vs. platinum-pemetrexed chemotherapy in the first-line setting. The study met its primary endpoint with a mOS of 18.1 months vs. 14.1 months (HR 95% CI, 0.73 [0.61–0.87]). The benefit of immunotherapy was found regardless of programmed death-ligand 1 (PD-L1) expression and regardless of histological subtypes, although the benefit was greater in patients with non-epithelioid histology. In fact, while mOS in non-epithelioid and epithelioid subtypes treated with immunotherapy were similar (18.1 vs. 18.2 months), mOS in patients treated with chemotherapy differed between non-epithelioid and epithelioid (8.8 months vs. 16.7 months, respectively). Thus, the positive result of the trial is in part driven by the great benefit of immunotherapy in patients with non-epithelioid histology. Furthermore, CheckMate 743 provided interesting data on durable responses: at 3 years, 28% of patients in the nivolumab plus ipilimumab group had ongoing responses, vs. 0% in the chemotherapy group. Safety was acceptable, with comparable percentages of all grade 3–4 treatment-related adverse events in both arms of the trial.

Following the results of Checkmate 743, the combination of Nivolumab and Ipilimumab has become a viable first-line option, to be especially preferred in non-epithelioid histologies. Moreover, other than on its own, the use of immunotherapy drugs has been investigated in combination with standard chemotherapy in a series of trials.

The anti-PD-L1 monoclonal antibody, Durvalumab, has been investigated in combination with platinum-pemetrexed as first-line treatment in two different phase-2 trials: DREAM and PrECOG 0505. In the single-arm DREAM trial [11] patients received a combination of chemotherapy plus Durvalumab for a maximum of six cycles, followed by maintenance with Durvalumab for a maximum of 12 months. In total, 57% of patients were alive and progression-free at 6 months, meeting the primary endpoint of the trial. ORR was 48% and mOS was 18.4 months. Similarly, the PrECOG 0505 trial [12] also met its primary endpoint, with a mOS of 20.4 months vs. 12.1 in the control arm with cisplatin-pemetrexed. The estimated percentages of progression-free patients at 6, 12, and 24 months were 67.3%, 18.2%, and 6.1%, respectively, and ORR was 56.4%. Given these promising results, this strategy is currently under investigation in the randomized phase 3 trial DREAM3R (NCT04334759) [13].

Other two phase 3 trials evaluating chemotherapy and immune checkpoint inhibitors are ongoing: the IND227 trial is examining the combination of anti-PD-1 Pembrolizumab and platinum-pemetrexed chemotherapy (NCT02784171) [14], while the ETOP BEAT-meso trial is evaluating the addition of anti-PD-L1 Atezolizumab to carboplatin/pemetrexed/bevacizumab (NCT0376201) [15].

### 2.3. Antiangiogenic

Malignant mesothelioma cell lines express elevated levels of both VEGF (vascular endothelial growth factor) and the VEGF receptors, VEGFR-1 (Flt-1) and VEGFR-2 (KDR), compared to normal mesothelial cells, and a correlation between VEGF serum levels and OS has been observed in MPM patients [16,17]. Thus, several trials have tried to target this important signaling pathway.

The randomized phase 3 MAPS trial [17] evaluated the addition of bevacizumab, a VEGF antibody, to cisplatin-pemetrexed therapy in 448 treatment naïve patients. The primary outcome of the trial, mOS, was positive (18.8 vs. 16.1 months, *p* = 0.0167), although a higher percentage of adverse events, related to the antiangiogenic drug, was noted. Around 24.3% of patients had to stop treatment due to toxicities, compared to 6% in the standard group. After the initial 4–6 cycles of cisplatin-pemetrexed + anti-VEGF, bevacizumab was continued until disease progression: this trial is the first positive evidence for maintenance therapy with an anti-VEGF agent [17].

Another antiangiogenic agent investigated in the first-line setting is Nintedanib, an oral triple angiokinase inhibitor targeting VEGF receptors 1–3, PDGF receptors α and β, FGF receptors 1–3, and Src and Abl kinases. Despite initial promising data, the phase 3 trial LUME Meso investigating the addition of Nintedanib to cisplatin-pemetrexed failed to reach its primary endpoint, PFS [18].

## 3. Novel Approaches in First-Line Therapy

### 3.1. Vaccines

The use of vaccines has been explored in different settings within the first-line realm: as first-line treatment in conjunction with chemotherapy and as maintenance following chemotherapy.

The CRS-207 vaccine is a weakened Listeria monocytogenes strain capable of stimulating the immune system. It has been investigated in a phase 1b trial [19], followed by pemetrexed and cisplatin chemotherapy, obtaining a DCR of 89% and ORR of 57%. Of note, tumor reduction was observed post-vaccine and prior to chemotherapy in 31% of patients. Similarly, since mesothelioma cancer cells often have an abundant expression of WT1 (Wilms’ tumor suppressor gene), a trial using dendritic cells (DC) targeting WT1, in conjunction with chemotherapy (platinum/pemetrexed), is ongoing [20]. This idea was based on initial trials on pretreated mesothelioma patients (although with different types of vaccines) that proved the activation of CD4+ and CD8+ T cells and initial clinical response [21]. 

Another strategy under study is the use of a maintenance vaccine after first-line chemotherapy. Several trials have used autologous dendritic cells loaded with tumor cell lysate [22,23,24]. A phase I trial using dendritic cells plus low dose cyclophosphamide after chemotherapy +/− surgery proved an acceptable safety and obtained a DCR in 8 out of 10 patients, although patients mostly exhibited early-stage disease [23]; another phase 1 trial obtained a DCR in 4/10 patients, with 1 partial response [24]. A phase 2/3 trial is ongoing [25]. 

### 3.2. Metabolism-Based Strategies

The idea of targeting the unique metabolism of cancer cells is being explored in several ways in mesothelioma patients, mostly but not exclusively in the first-line setting. Arginine is a semi-essential amino acid involved in tumor growth, which can be synthesized by the enzyme argininosuccinate synthase 1 (ASS1). Intratumoral deficiency of ASS1 has been detected in a significant number of patients with cancers, including mesothelioma [26]. Tumors with ASS1 loss are unable to synthesize arginine and they depend on extracellular arginine for survival. Thus, a promising therapeutic strategy may involve the depletion of systemic arginine by using pegylated arginine deiminase (ADI-PEG). In the phase 2 ADAM trial, patients with ASS1-deficient MPM, chemotherapy naïve or progressing to first-line chemotherapy, were randomized to receive ADI-PEG20 plus best supportive care or best supportive care alone. Patients who received ADI-PEG20 had a significantly longer mPFS of 3.2 months vs. 2.0 months of the control group [27]. Promising results were also seen in the phase 1 TRAP study, in which ADI-PEG was added to cisplatin-pemetrexed in treatment naïve patients with ASS1-deficient MPM [28]. This strategy is also currently under investigation in the phase 2/3, randomized, double-blind, ATOMIC trial, involving patients with sarcomatoid or biphasic mesothelioma (NCT02709512) [29].

Another strategy involves targeting the production of adenosine triphosphate (ATP), which is normally achieved through purine synthesis or the adenine salvage pathway, involving the breakdown of methylthioadenosine (MTA) by the enzyme methylthioadenosine phosphorylase (MTAP). The loss of MTAP has been reported in a variety of solid tumors, including mesothelioma [30]. Thus, MTAP-deficient tumors are dependent on the de novo purine synthesis pathway, which can be inhibited by several drugs. One of these drugs is L-alanosina, which has been evaluated in a phase 2 trial including 65 patients with MTAP-deficient solid tumors (16 mesotheliomas): there were no objective responses, although 24% had stable disease, including 2 patients with mesothelioma [31]. Another strategy to exploit this metabolic pathway is being evaluated in a phase ½ study, utilizing MRTX1719, a potent PRMT5-MTA inhibitor in pretreated patients with MTAP-deficient solid tumors (NCT05245500) [32].

### 3.3. Hsp90 Inhibitor

Heat shock protein 90 (Hsp90) is a chaperone that allows the correct maturation and stability of a variety of proteins. Its inhibition leads to the degradation of its client proteins, thus impairing the growth and survival of cancer cells [33]. MESO-02 is a phase 1/2 trial of first-line Ganetespib, an Hsp90 inhibitor, with pemetrexed-platinum in patients with MPM. This study was not powered to detect improvements in efficacy but the results were encouraging. Overall, partial response was observed in 14 out of 27 patients (52%) and DCR was 81%, and for patients treated at the maximum tolerated dose of Ganetespib, 10 out of 18 patients (56%) had a partial response and all had disease control (100%) [34]. No other trials are ongoing. 

## 4. Conclusions on First-Line Therapy

At the moment the landscape of first-line therapy for advanced mesothelioma sees two viable options: cisplatin-pemetrexed doublet and immunotherapy combination with Nivolumab and Ipilimumab. Both options can be considered in the presence of epithelioid mesothelioma, while the results of Checkmate 743 clearly point to better efficacy of immunotherapy in the setting of non-epithelioid histological subtypes. Given the efficacy of these two strategies, the idea of combining both chemotherapy and immunotherapy seems promising. Considering the positive phase 2 trials, the results of the ongoing phase 3 trials investigating several combinations (platinum-pemetrexed plus Durvalumab/Pembrolizumab/Atezolizumab-Bevacizumab) are eagerly awaited. 

A similar idea involves the use of vaccines to stimulate the immune system, again in association with standard chemotherapy. While interesting, it should be noted that data regarding this technique are still in a premature phase, as they come mostly from phase 1 trials, and more robust data are needed from ongoing trials.

Another idea for combination therapy with standard chemotherapy involves Hsp90 inhibitor Ganetespib, with interesting results in phase 1/2 trials. However, this agent should be better investigated in a larger trial but none are currently ongoing, to our knowledge.

The metabolic route to achieve tumor suppression is in a slightly more advanced stage. Although targeted only to those patients with ASS1 loss, promising results come from the phase 2 ADAM trial on ADI-PEG20, with an ongoing randomized phase 2/3 trial (ATOMIC). On the other hand, for MTAP deficient mesothelioma, although L-alanosina is not being further investigated, at least one trial is ongoing with a different drug, PRMT5-MTA inhibitor.

To conclude, the implementation of anti-angiogenic drugs has been disappointing. The use of Nintedanib has not been associated with an increase in OS and Bevacizumab, despite having a formally positive phase 3 trial, is plagued by a high incidence of adverse events requiring treatment interruption and a relatively low gain in OS, and it is not commonly used in clinical practice (Table 1).

## 5. Second-Line Treatments

### 5.1. Immunotherapy

Drawing from the promising results of two phase 2 trials [35,36], the phase 3 CONFIRM [37] trial involved 332 pretreated patients with pleural or peritoneal mesothelioma, who were randomized to receive Nivolumab 240 mg or placebo. The study was positive both in PFS value (3.0 vs. 1.8 months, *p* = 0.0012) and mOS (10.2 vs. 6.9 months, *p =* 0.0090). Interestingly, the disease control rate for Nivolumab was 64% vs. 50% for the placebo arm, with rates of stable disease being, respectively, 53% vs. 49%. 

Inconsistent with the results of other trials involving immunotherapy (such as CheckMate 743), subgroup analysis showed that epithelioid mesothelioma was the only type associated with a meaningful hazard ratio for both PFS (0.64 (0.50–0.83)) and OS (0.67 (0.50–0.91)), while in non-epithelioid subtypes the advantage was more dubious (HR for PFS 0.77 (0.37–1.60) and HR for OS 0.79 (0.35–1.80)). This could be explained by the fewer number of patients with non-epithelioid mesothelioma involved (39 patients overall) and the partial number of events at the time of the study publication (25/39); however, the benefit of immunotherapy in the second line according to histotypes remains an open question. 

Furthermore, it should be noted that real-world data from a Dutch expanded access program [38] for Nivolumab in second or further lines of therapy showed slightly worse results: PFS was 2.3 months and OS was 6.7 months, DCR 37%, and ORR 10%.

Interestingly, despite promising results from earlier papers [39,40], the phase 3 PROMISE- meso trial [5] comparing Pembrolizumab vs. gemcitabine or vinorelbine, failed to prove an advantage in PFS (HR 1.06, *p* = 0.76) or OS (HR 1.04, *p =* 0.85), even when adjusting for crossover, although ORR was better in the experimental arm (22% vs. 6%, *p* = 0.004). Discouraging results from immune-monotherapy also come from the phase 2b trial DETERMINE [41], involving the anti-CTLA-4 Tremelimumab vs. placebo. The trial failed to reach a benefit in OS, its primary endpoint, despite positive results from a previous single-arm trial, the MESOT-TREM-2012 [42]. Another anti-PD-L1 drug, Avelumab, was investigated in a phase 1b trial (JAVELIN Solid Tumor Trial) [43], reaching an ORR of 9% (5/53 patients, including one complete response) and DCR of 58%. However, no other trials are ongoing with this drug.

Interesting results come from two phase 2 trials investigating combination immunotherapy with anti-PD-(L)1 + anti CTLA4. The phase 2, non-comparative, IFCT-1501 MAPS2 trial [44] explored the possibility of administering Nivolumab-Ipilimumab or Nivolumab monotherapy. DCR at 12 weeks was 40% for the arm receiving Nivolumab monotherapy and 52% in the combination arm; ORR was 20% and 35%, respectively, and mOS was 11.9 vs. 15.9 months. Worse adverse events were registered in the combination arm: grade 3–4 events were seen in 14 vs. 26% of patients, and three grade 5 events were reported in the combination arm. The combination of Tremelimumab and Durvalumab was explored in the single-arm phase 2 NIBIT-MESO-1 trial [45], involving patients in 1st or 2nd line (30% and 70%, respectively). It achieved its primary endpoint of immune response rate, with 28% responding (duration of response 16.1 months) and 65% achieving DCR, with an acceptable safety profile. Looking exclusively at the subset of patients in the second line setting (28 patients), similar results can be seen for ORR (28%) and DCR (68%), with an impressive OS of 16.6 months. 

It should be noted that in all trials involving immunotherapy in this setting no biomarker has been proven to correlate with response, as data on PD-L1 continue to be inconsistent. Thus, at the moment, clinicians have no reliable way to select patients likely to benefit from this type of treatment.

### 5.2. Chemotherapy in Second Line Setting

Several chemotherapy agents have been evaluated during the years as a second-line strategy. Recently, the updated results of the phase 2 VIM trial [46], investigating the use of oral vinorelbine against active symptom control, showed an advantage in PFS (4.2 vs. 2.8 months, *p =* 0.002), but not in OS (9.3 vs. 9.1 months, *p* not provided) or DCR (65.3 % vs. 48.2 %, *p =* 0.06). Similar results for intravenous vinorelbine have been shown in a retrospective work [47]. One more paper [48] retrospectively analyzed the use of vinorelbine or gemcitabine in 60 pretreated patients, obtaining a DCR of around 50% but with dubious efficacy on PFS (1.7 for vinorelbine and 1.6 for gemcitabine) and OS (5.4 and 4.9 months, respectively).

Another investigated chemotherapy regimen was pemetrexed monotherapy. A phase 3 trial [49] in pemetrexed naïve patients showed an improvement in PFS but not in OS when compared to the best supportive care. It is worth noting that patients showing an objective response to pemetrexed (18.7%) had a prolonged OS of 20.5 months. However, given the use of cisplatin-pemetrexed doublet as first-line therapy, today, the possible role of pemetrexed monotherapy could be in those patients ineligible for a platinum-based doublet or progressing to vinorelbine.

Furthermore, the use of a histone deacetylase inhibitor, vorinostat, was investigated in the phase 3 trial VANTAGE-014, where it failed to reach a statistically significant benefit in OS compared to placebo, although PFS was improved (6.3 vs. 6.1 months, *p* < 0.001) [50].

Regarding the use of combination chemotherapy, a small Japanese retrospective paper [51] and an Italian phase 2 trial [52] investigated the use of vinorelbine plus gemcitabine, with modest activity but manageable toxicity. The combo oxaliplatin with or without gemcitabine was also analyzed in a small paper in second or further lines of therapy with analogous results [53]. 

## 6. Novel Approaches in Second and Further Lines of Therapy

### 6.1. New Types of Immunotherapy

Regarding immunotherapy, other than anti-PD-L1/PD-1 agents, several new drugs directed against less-known targets are being investigated: CA-170 is a novel oral immune checkpoint inhibitor targeting both PD-L1/PD-L2 and VISTA (V-domain Ig suppressor of T cell activation). It is being examined in a phase 1 trial for solid tumors including mesothelioma (NCT02812875) [54].INCAGN01949 is an anti-OX40 agonist human antibody. It has been studied in a dose-finding phase 1/2 trial (NCT02923349) that included 87 patients with advanced solid tumors (3 mesothelioma cases) obtaining a DCR of 27.6% [55]. Another trial testing the combination of INCAGN01949 plus Ipilimumab and Nivolumab was discontinued due to limited clinical activity [56].INCAGN02385 is an anti-LAG-3 (Lymphocyte activation gene-3) drug under study in a phase 1 trial including several malignancies (NCT03538028) [57].INCAGN02390 is a new drug directed against TIM-3 (T-cell immunoglobulin and mucin domain-containing protein 3). A dose-finding phase 1 trial [58] recently showed a preliminary DCR of 17.5%. However, no specific data on mesothelioma were available, and the broad range of malignancies and previous therapies received (58% had immunotherapy previously) points to the need for further trials.

A different take on immunotherapy involves the use of vaccines to stimulate the immune system, mostly involving administration through infusion in the pleural space with a catheter, as opposed to the standard oral or intravenous method for drug delivery. There were several phase 1 trials investigating the use of gene therapy in the form of an adenoviral vector containing the human interferon-beta gene (BG00001), designed to be used in both pleural mesothelioma and pleural effusion due to other malignancies [59,60,61]. Published results point to a relatively tolerable drug, capable of eliciting an immune and associated with some long-lasting stable disease and response to PET assessment. However, due to the company’s decisions, the trials were stopped [60,61]. 

Another phase 1 trial involved the administration of an adenoviral vector containing the human interferon-alpha2b gene (Ad.hIFN-α2b) with concomitant celecoxib, followed by chemotherapy, both in first- and second-line settings. The trial had interesting results with an ORR of 25% and DCR of 88%, mOS in the first-line cohort was 12 months and 17 months in the second-line cohort, with a strong difference between pemetrexed based (26 months) and gemcitabine-based (10 months) second line [62]. Furthermore, the combination was well tolerated, with most toxicities being related to cytokine release syndrome secondary to the initial vector dose. More recent ongoing trials on vaccines in this setting involve combination therapies as well, such as an ongoing phase 3 trial [63] analyzing the use of intrapleural Adenovirus-Delivered Interferon Alpha-2b (rAd-IFN) combined with oral Celecoxib and intravenous Gemcitabine. 

There are also preliminary results from a phase 1/2 trial involving standard-of-care therapy plus a granulocyte-macrophage colony-stimulating factor expressing oncolytic adenovirus ONCOS-102 (Ad5/3-D24-GMCSF) vs. placebo in first- or second-line patients [64]. Although an ORR and DCR advantage was dubious, OS at 12 months in first-line was 64% vs. 50%. Other trials are exploring the use of combination therapies with anti-PD-L1/PD-1 inhibitors. There is an ongoing trial with Nivolumab associated with a WT1 vaccine called Galinpepimut-S [65] and another with Nivolumab associated with intratumoral injection of a replication-incompetent adenovirus into which the gene for REIC/Dkk-3 has been inserted (MTG201) [66]. Another promising trial sees the use of Pembrolizumab combined with a dendritic cell intradermal vaccine in PD-L1 negative pretreated patients (MESOVAX trial) [67].

### 6.2. Antiangiogenic

In parallel with the MAPS trial in the first-line setting, other trials have explored the use of antiangiogenic drugs in pretreated patients, usually within a combination regimen. The phase 2 RAMES trial [68] saw an improvement in OS for the combination of Gemcitabine-Ramucirumab vs. Gemcitabine-placebo in patients pretreated with platinum-pemetrexed, reaching 13.8 vs. 7.5 months (HR 0.71, *p =* 0.028), with even greater advantage for non-epithelioid subtypes (HR 0.37). Despite a low ORR, DCR was obtained in 73% vs. 52% of patients. Another phase 2 trial investigating the use of ramucirumab plus Nivolumab is ongoing [69]. 

Other antiangiogenics had worse performances. Bevacizumab has been investigated alongside anti-EGFR Erlotinib with poor results [70]. Furthermore, in a phase 1b basket trial (PEMBIT) analyzing Pembrolizumab plus the oral antiangiogenetic Nintedanib, although some types of cancer showed a response to this combination, this was not observed in the two mesothelioma cases included [71].

### 6.3. Mesothelin

Mesothelin (MSLN) is a glycoprotein expressed in the mesothelial cells of the pleura, peritoneum, and pericardium and it is reported to be highly expressed in several types of malignant tumors, including mesothelioma. Epithelioid mesotheliomas exhibit reactivity for mesothelin, but this was not noted in sarcomatoid variants or in the spindle cell component of biphasic mesothelioma [72,73,74]. Low expression in normal human tissues and high expression in many cancers makes it an attractive target for therapy. 

Amatuximab is a mouse-human chimeric anti-mesothelin antibody. In a phase 2 study, it was added to the standard first line with cisplatin and pemetrexed, with a promising mOS of 14.8 months [75]. Unfortunately, a subsequent randomized phase 2 trial (ARTEMIS) with amatuximab was prematurely closed to low accrual (NCT02357147) [76]. 

Subsequent trials explored the use of anti-mesothelin drugs in the second-line setting. For example, anetumab-ravtansine is a human anti-mesothelin antibody conjugate to the maytansinoid tubulin inhibitor DM4. Despite the encouraging preliminary results on activity in a phase 1 trial [77], a randomized phase 2 study (NCT02610140) [78] comparing anetumab-ravtansine vs. vinorelbine in second-line treatment in MSLN-positive MPM, failed to show an improvement in PFS. However, a phase 1/2 trial is ongoing, evaluating anetumab-ravtasine with pembrolizumab in MSLN-positive pretreated patients (NCT03126630) [79]. 

SS1P is an anti-mesothelin immunotoxin composed of a targeting antibody fragment fused to a fragment of Pseudomonas exotoxin A. In a phase 1 trial, SS1P show little activity because of generated neutralizing antibodies to the pseudomonas toxin [80]. To avoid this problem, a subsequent phase 1 trial utilized the administration of upfront chemotherapy to deplete T and B lymphocytes and reduce the development of neutralizing antibodies: the subsequent administration of SSP1 was associated with partial response in 3/10 patients [81]. 

Another promising approach is the use of anti-MSLN Chimeric Antigen Receptor (CAR) T-cell therapy directed toward tumor antigens, such as mesothelin. Multiple phase 1–2 clinical trials investigated anti-MSLN CAR-T, alone or in association with checkpoint inhibitors, with or without prior lymphodepletion. A phase 1 trial evaluating lentiviral-transduced CAR-T cells anti-MSLN in patients with pretreated mesothelioma, pancreatic and ovarian cancer demonstrated stable disease in 11 of 15 patients, with good tolerance [82]. In a multi-arms phase 1, fully human anti-MSLN CAR-T cells were administered intrapleurally in pretreated patients with primary or secondary pleural malignancies (25/27 with mesothelioma). mOS in mesothelioma patients who received CAR-T therapy was 17.7 months, while in patients who received CAR-T therapy plus Pembrolizumab OS was 23.9 months [83]. Possible limiting factors in the use of CAR-T cell therapy are the heterogenous antigen expression, the difficulty in obtaining tumor infiltration from T-cells, and the presence of an immunosuppressive microenvironment that inhibits the function of CAR T-cell [84,85]. Furthermore, safety concerns have been raised due to episodes of anaphylaxis in clinical trials (NCT01355965) [86].

### 6.4. Other Targets

Numerous new targets have been tested in the setting of advanced mesothelioma.

Anti-EGFR (epidermal growth factor receptor) drugs such as Gefitinib [87] and Erlotinib [88] did not show any benefit, despite the presence of EGFR hyperexpression.Focal adhesion kinase (FAK) is a non-receptor tyrosine kinase with an important role in cancer cell survival and immune system evasion [89]. Defactinib (Vs-6063) is an anti-FAK drug that is being evaluated in combination with other drugs in the relapsed setting. A trial in combination with Pembrolizumab is ongoing [90], while a trial with a combination of anti-FAK and a dual PI3K/mTOR inhibitor was terminated in 2015 (although no results are available) [91]. Defactinib’s efficacy was also assessed as maintenance therapy after response or stability to first-line chemotherapy but it failed to show any gain in OS or PFS [92]. Other trials involve the multi-tyrosine kinase inhibitor APG-2449 (anti-FAK, ALK, and ROS1) in an ongoing phase 1 trial [93], and the combination of the anti-FAK GSK2256098 and anti-MEK Trametinib in a dose-finding phase 1b trial that showed limited efficacy with a PFS of 2.6 months and no radiological responses [94].Hyperexpression of EZH2 (enhancer of zeste homolog), a histone-lysine N-methyltransferase, is often found in association with BAP1 loss and it has been implicated with epigenetic regulation and oncogene functions. Its inhibitor, Tazemetostat, has been used in relapsed mesothelioma patients in a phase 2 trial with a favorable toxicity profile and over 50% disease control rate at 12 weeks (primary endpoint of the trial); furthermore, 28% of patients had sustained disease control at 24 weeks [95].BRCA1-associated protein 1 (BAP1) mutations are found in several aggressive cancers, including malignant mesothelioma. One of the roles of BAP1 is to regulate homologous recombination DNA damage repair, suggesting that targeting this pathway with PARP inhibitors may have therapeutic value. Mesothelioma Stratified Therapy (MiST) is a stratified multi-arm phase 2a clinical trial to enable accelerated evaluation of targeted therapies for relapsed malignant mesothelioma. Patients with BAP1-deficient or *BRCA1*-deficient mesothelioma are stratified to receive the PARP inhibitor Rucaparib. The disease control rate at 12 weeks was 58% and at 24 weeks was 23% [96].AXL, a member of the TAM family tyrosine kinase receptors, is overexpressed in 74% of mesothelioma and in cells of the tumor microenvironment, where it has an important role in immune evasion: the oral inhibitor Bemcentinib is being evaluated in combination with Pembrolizumab in one arm of the previously cited MiST trial [97].Often expressed in mesothelioma, CDKN2A regulates the cell cycle by inhibiting the tumor suppressor p16ink4A, an endogenous suppressor of cyclin-dependent kinase (CDK) 4 and CDK6. Arm 2 of the MiST trial is investigating Abemaciblib, a CDK4/6 inhibitor, in p16ink4A-negative pretreated mesothelioma. The disease control rate was 54% at 12 weeks [98].

## 7. Conclusions on Second-Line Therapy

These recent years saw the introduction of immunotherapy in the treatment paradigm for mesothelioma. Trials such as the phase 3 CONFIRM trial, and to a lesser extent smaller phase 2 trials, suggest an increased OS advantage in using Nivolumab in the second or further line of therapy. This advantage seems to be driven only in small part by objective responses, but rather by an increased disease control rate and by a small subset of patients with long-lasting responses. Furthermore, none of the trials involving Nivolumab were compared to a strong second-line option such as vinorelbine or gemcitabine, so the real-life impact is still nebulous and should require further testing, especially considering that the only trial comparing immunotherapy (albeit with Pembrolizumab and not Nivolumab) against chemotherapy, did not show any benefit. 

Promising results have been observed with combination treatment with antiCTLA-4 and PD-L1 (Nivolumab-Ipilimumab and Tremelimumab-Durvalumab), although again with no real active comparison arm and with a small subset of patients. Considering the results obtained in the first-line setting with an immunotherapy combination, it is reasonable to assume that this road warrants further trials in the second-line setting as well. 

Regarding the use of chemotherapy in the second line, the VIM trial is one of the few papers that proved an advantage in PFS vs. active symptom control. However, given that the patients included had only previously received platinum-based chemotherapy, the role of vinorelbine today is still unclear in the subset of patients progressing after first-line immunotherapy. It should be noted that there is no proven benefit for other types of chemotherapy, either monotherapy or combination, as the data available are mostly derived from retrospective papers or small single-arm phase 2 trials.

Regarding the role of new types of immune therapies, many trials are exploring how to stimulate the immune system using different targets. The trials on new immune checkpoint targets such as anti-VISTA, OX40, LAG-3, and TIM-3 are all in their infancy, with data coming from basket trials or ongoing phase 1 trials, thus more data on mesothelioma patients specifically are needed. Similarly, despite being investigated in many trials, the use of vaccines is still in the early stages, possibly due to the many types of vaccines that have been analyzed over the years with inconsistent results. The recent preliminary results on the use of vaccines in combination with either chemotherapy or other types of immunotherapy are promising and several trials are ongoing.

Despite the good results of the MAPS trial in the first line with Bevacizumab and chemotherapy, the only antiangiogenic drug that showed promising results in the second line setting is Ramucirumab in association with gemcitabine. However, it should be noted that the quality of evidence for the use of this combination is not strong, as it comes from a phase 2 trial. A trial with the combination Ramucirumab-Nivolumab is ongoing. 

Regarding the use of mesothelin as a target, two interesting lines are being explored, one with the drug conjugate anetumab-ravtansine and the other with anti-mesothelin CAR-T therapy. Anetumab-ravtansine had mixed results from different phase 1 trials, so the results of a phase 1/2 trial with pembrolizumab are needed to better evaluate its efficacy. Regarding CAR-T therapy, despite having some initial promising results, all the ongoing trials are in the early stages so it is unlikely that we will have confident results on its efficacy soon. 

Several TKI are under evaluation. So far, the results of the phase 2 trial involving anti-EZH2 Tazemetostat are promising, and the results of the MiST trial are also eagerly awaited, especially given the preliminary results on PARP inhibitor Rucaparib and anti-CDK4/6 inhibitor Abemaciclib. It should be once again noted that a phase 3 trial is not underway for any TKI. 

To conclude, there is a strong need for large phase 3 trials comparing immunotherapy against an active arm such as vinorelbine, ideally including a stratification between histological subtypes, with the aim to assert the real benefit of immunotherapy in the second-line setting and identify new predictors of response. An open question remains whether the trials should focus on mono immunotherapy or a combination of anti-PD-(L)1 plus anti-CTLA4; with the exception of Nivolumab, standing the disappointing results of other trials involving mono immunotherapy, we feel that trials with combination immunotherapy represent a promising route that should be explored. With respect to the novel therapies in the second line setting of advanced mesothelioma, many lines of research are being explored, but unfortunately, none of them are in the advanced stage of testing. Standing the numerous concluded (Table 2) and ongoing trials (Table 3), we hope that the treatment scenario for advanced mesothelioma will soon evolve to include more options.

**Table 1 ijms-24-10415-t001:** First-line trials with corresponding results. When available, HR and 95% confidence intervals are provided. PFS: Progression Free Survival; OS: Overall Survival; ORR: Overall Response Rate; DCR: Disease Control Rate; ITT: intention to treat population.

	Trial	Phase of Trial	Enrolled Patients	Experimental Arm	Standard Arm/ Placebo/ Single Arm	PFS (Months)	OS (Months)	ORR	DCR	NOTES
**Chemotherapy**	EMPHACIS [4]	3	456	Pemetrexed/cisplatin	Cisplatin	5.7 vs. 3.9	12.1 vs. 9.3 [95% CI 10.0–14.4 vs. 7.8–10.7, HR 0.77]	41.3 vs. 16.7% [95% CI 34.8–48.1 vs. 12.0–22.2]		
	Jan P. van Meerbeeck et al. [5]	3	250	Ralitrexed/cisplatin	Cisplatin	5.3 vs. 4	11.4 vs. 8.8 [HR 0.76, 95% CI 0.58–1.00]	23.6 vs. 13.6%	76% vs. 67.9%	
	CALGB 30901 [8]	2	49	Pemetrexed maintenance	Placebo	3.4 vs. 3 [95% CI 2.8–9.8 vs. 2.6–11.9, HR 0.99]	16.3 vs. 11.8 [95% CI 105–26.0 vs. 9.3–28.7, HR 0.86]	11.1 vs. 0%	55.5 vs. 66.6%	
	NVALT19 [9]	2	130	Gemcitabine maintenance	BSC	6.2 vs. 3.2 [95% CI 4.6–8.7 vs. 2.8–4.1, HR 0.48]	16.4 vs. 13.4 [95% CI, 11.6–20.2 vs. 12.4–17.8, HR 0.90]	17 vs. 4%		
**Immunotherapy (anti PD-1/PD-L1)**	Checkmate 743 [10]	3	605	Nivolumab/ipilimumab	Pemetrexed/cis-carboplatin	6.8 vs. 7.2 [95% CI 5.6–7.4 vs. 6.9–8.0, HR 0.92]	18.1 vs. 14.1 [95% CI 16.8–21.0 vs. 12.4–16.3, HR 0.73]	40 vs. 43%	77 vs. 85%	
	DREAM [11]	2	54	Pemetrexed/cisplatin/ durvalumab	Single arm	6.9 [95% CI 5.5–9.0]	18.4 [95% CI 13.1–24.8]	48%	87%	
	PrECOG 0505 [12]	2	55	Pemetrexed/Platin bases/durvalumab	Single arm	6.7 [95% CI 6.1–8.4]	20.4 [95% CI 13.0–28.5]	56.4%	92.7%	
**Angiogenetic**	MAPS trial [17]	3	448	Bevacizumab/ pemetrexed/cisplatin	Pemetrexed/ cisplatin	9.2 vs. 7.3 [95% CI 8.5–10.5 vs. 6.7–8.0, HR 0.61]	18.8 vs. 16.1 [95% CI 15.9–22.6 vs. 14.0–17.9, HR 0.77]			
	LUME-Meso [18]	3	458	Pemetrexed/cisplatin/ nintedanib	Pemetrexed/ cisplatin/ placebo	6.8 vs. 7 [95% CI 6.1–7.0 vs. 6.7–7.2, HR 1.01]	14.4 vs. 16.1 [95% CI 12.2–17.9 vs. 13.7–19.3, HR 1.12]	45 vs. 43%	91 vs. 93%	
**Vaccine**	Hassan et al. [19]	1b	35	CRS-207 + pemetrexed/cisplatin	Single arm	7.5 [95% CI 7.0–9.9]	14.7 [95% CI 11.2–21.9]	57%	89%	
	Cornelissen et al. [23]	1	10	Dendritic cells	Single arm		33.8	10%	80%	5/10 patients had surgery following chemotherapy and then received the vaccine
	Hegmans et al. [24]	1	10	Dendritic Cells	Single arm		19 [95% CI 11–34]	10%	40%	
**Metabolism-based**	ADAM trial [27]	2	68	ADI-PEG20/BSC	BSC	3.2 vs. 2 [HR 0.56, 95% CI 0.33–0.96]	11.5 vs. 11.1 [HR 0.68, 95% CI 0.39–1.16]			The trial included 1st and 2nd line patients
	TRAP [28]	1	9	ADI-PEG 20/pemetrexed/cisplatin	Single arm	7.5	13.9	78% [95% CI 39–97%]		
	Kindler et al. [31]	2	65	L-alanosine	Single arm			0%	24%	The trial included 1st and 2nd line patients
**HSP90 Inhibitor**	MESO-02 [34]	1b	27	Ganetespib/pemetrexed/ cis or carboplatin	Single arm	5.8 [95% CI 5.0–80]	11.5 [95% CI 8.0–19.5]	52% [95% CI 32–71%]	81%	
**Mesothelin**	Hassan et al. [75]	2	89	Amatuximab/pemetrexed/cisplatin	Single arm	6.1 [95% CI 5.8–6.4]	14.8 [95% CI 12.4–18.5]	39.8% [95% CI 29.2–51.1%]	90%	
**TKI**	NCT01870609 (COMMAND) [92]	2	344	Defactinib (Anti FAK)	Placebo	4.1 vs. 4.0 [95% CI 2.9–5.6 vs. 2.9–4.2]	12.7 vs. 13.6 [95% CI 9.1–21.0 vs. 9.6–21.2, HR 1.0]	4.0 vs. 2.9%	62.4 vs. 63.7%	Study was terminated at first interim analysis for futility

## Figures and Tables

**Table 2 ijms-24-10415-t002:** Second or further line trials, with corresponding results. When available, HR and 95% confidence intervals are provided. PFS: Progression Free Survival; OS: Overall Survival; ORR: Overall Response Rate; DCR: Disease Control Rate; ITT: intention to treat population.

	Trial	Phase/ Type of Trial	Enrolled Patients	Experimental Arm	Standard Arm/ Placebo/ Single Arm	PFS (Months)	OS (Months)	ORR	DCR	NOTES
**Immunotherapy** **(PD-L1/PD-1)**	Quispel-Janssen J et al. [35]	2	34	Nivolumab	Single arm	2.6 [95% CI 2.2–5.5]	11.8 [95% CI 9.7–15.7]	24% at 12 weeks	47% at 12 weeks	
	MERIT [36]	2	34	Nivolumab	Single arm	6.1 [95% CI 2.9–9.9]	17.3 [95% CI, 11.5–not reached]	29% [95% CI 16.8–46.2]	68%	
	CONFIRM [37]	3	332	Nivolumab	Placebo	3 vs. 1.8 [95% CI, 2.8–4.1 vs. 1.4–2.6, HR 0.77]	10.2 vs. 6.9 [95% CI 8.5–12.1 vs. 5.0–8.0, HR 0.69]	11 vs. 1%	64 vs. 50%	
	Cantini et al. [38]	Retrospective	107	Nivolumab	Single arm	2.3	6.7	10% at 12 weeks	37% at 12 weeks	
	KEYNOTE 028 [39]	1b	25	Pembrolizumab	Single arm	5.4 [95% CI 3.4–7.5]	18.0 [95% CI 9.4-not reached]	20% [95% CI 6.8–40.7]	72%	
	KEYNOTE 158 [40]	2	118	Pembrolizumab	Single arm	2.1 [95% CI 2.1–3.9]	10 [95% CI 7.6–13.4]	8% [95% CI, 4–15% ]	45%	
	DETERMINE [41]	2b	658, ITT 571	Tremelimumab	Placebo	-	7.7 vs. 7.3 [95% CI 6.8–8.9 vs. 5.9–8.7, HR 0.92]	4.5 vs. 1.1% [95% CI 2.6–7.0 vs. 0.1–3.8]	27.7 vs. 21.7% [95% CI 23.3–32.5 vs. 16.0–28.3]	
	MESOT-TREM-2012 [42]	2	29	Tremelimumab	Single arm	6.2 [95% CI 5.7–6.7]	11.3 [95% CI 3.4–19.2]	7%	52%	
	JAVELIN Solid [43]	1b	53	Avelumab	Single arm	4.1 [95% CI 1.4–6.2]	10.7 [95% CI 6.4–20.2]	9% [95% CI 3.1–20.7]	58%	The trial included patients in 2nd or further lines of therapy (up to 8)
	IFCT-1501 MAPS2 [44]	2	125	Nivolumab + Ipilimumab	Nivolumab	5.6 vs. 4.0 [95% CI 3.1–8.3 vs. 2.8–5.7]	15.9 vs. 11.9 [95% CI 10.7– not reached vs. 6.7–17.7]	35 vs. 20%	52 vs. 40% at 12 weeks [95% CI 39–64 vs. 28–52]	Higher rates of adverse events were seen in the combination arm.
	NIBIT-MESO-1 [45]	2	40 (28 in 2nd line)	Tremelimumab + Durvalumab	Single arm	8 [95% CI 6.7–9.3] (8 months in 2nd line)	16.6 [ 95% CI 13.1–20.1] (16.6 months in 2nd line)	28 % [95% CI 15–44%] (25% in 2nd line)	65% [95% CI 48–79] (68% in 2nd line)	The trial included 1st and 2nd line patients.
**Chemotherapy**	VIM [46]	2	154	Oral vinorelbine + Active symptom control	Active symptom control	4.2 vs. 2.8 [95% CI 2.2–8.0 vs. 1.4–4.1, HR 0.6]	9.3 vs. 9.1	3% vs. 2%	65.3 vs. 48.2%	
	Zucali et al. [47]	Retrospective	59	Intravenous vinorelbine	Single arm	2.3 [95% CI 0.6–22.5]	6.2 [95% CI 0.8–27.8]	15.2%	49.1%	The trial included 2nd and 3rd line patients
	Zauderer et al. [48]	Retrospective	60	Vinorelbine or Gemcitabine	Single arm	1.7–1.6 [95% CI 1.3–2.9 and 1.3–3.6]	5.4–4.9 [95% CI 3.8–7.4 and 3.6–8.8]	0–4%	50–41%	The numbers are relative to vinorelbine and gemcitabine, respectively, also the trial included 2nd and 3rd line patients.
	Jassem et al. [49]	3	243	Pemetrexed	Best supportive care	3.6 vs. 1.5 [95% CI 3.0–4.4 vs. 1.5–1.9]	8.4 vs. 9.7 [95% CI 6.2–10.5 vs. 8.4–10.9]	18.7 vs. 1.7 %	59.3 vs. 19.2%	Patients had not previously received pemetrexed in 1st line
	VANTAGE 014 [50]	3	661	Vorinostat	Placebo	6.3 vs. 6.1 [95% CI 6.1–7.1 vs. 6.0–6.1, HR 0.75]	30.7 vs. 27.1 (weeks) [95% CI 26.7–36.1 vs. 23.1–31.9, HR 1.32]	1 vs. <1%	-	
	Toyokawa et al. [51]	Retrospective	17	Vinorelbine and Gemcitabine	Single arm	6.0	11.2	18% [95% CI 3.8–43.4%]	82%	The trial included 2nd and 3rd line patients
	Zucali et al. [52]	2	30	Vinorelbine and Gemcitabine	Single arm	2.8 [95% CI 0.6–12.1%]	10.9 [95% CI 0.8–25.3]	10%	43.3% [95% CI 25.5–62%]	
	Xanthoupoulos et al. [53]	Prospective observational	29	Oxaliplatin +/− Gemcitabine	Single arm	2.1	5.6 (95% CI 1.2–22.4)	6.9 %	44.8%	The trial included 2nd^,^ 3rd, 4th, 5th line patients
**Immunotherapy (targets other than PD-L1/PD1** **)**	NCT02923349 [55]	1/2	87	INCAGN01949	Single arm	-	-	1.1%	27.6%	
	NCT03241173 [56]	1/2	52	INCAGN01949 + Nivolumab + Ipilimumab	Single arm					Phase 2 (efficacy) was not carried on due to scarce benefit in phase 1 (dose finding)
	NCT03652077 [58]	1	40	INCAGN02390	Single arm	-	-	-	17.5%	Preliminary results. The number of patients with mesothelioma is unknown
**Vaccines**	Krug et al. [21]	1	9	WT1 vaccine	Single arm	-	-	0%	11%	3 patients with NSCLC were also included
	Sterman et al. [60]	1	17	Ad.hIFN-β (BG00001)	Single arm	-	-	5.9%	29–64%	Data are relative to CT and PET FDG assessment, respectively
	Sterman et al. [61]	1	10	Ad.hIFN-β (BG00001)	Single arm	-	-	0-/20%	40–50%	Data are relative to CT and PET FDG assessment, respectively
	Sterman et al. [62]	1	40	Ad.hIFN-alpha 2b (SCH 721015)	Single arm	-	13	25%	87.5%	The trial included 1st and 2nd line pt
	NCT02879669 [64]	½	31	Standard of care + Ad5/3-D24-GMCSF)	Standard of care	-	OS at 12 months in first line: 64 vs. 50	30–11% (1st–2nd line experimental) vs. 33–60% (1st–2nd line standard)	90–67% (1st–2nd line experimental) vs. 83–80% (1st -2nd line standard)	The trial included 1st and 2nd line patients. Preliminary results only
**Antiangiogenic**	RAMES [68]	2	161	Ramucirumab + Gemcitabine	Placebo + Gemcitabine	6.4 vs. 3.3 [70% CI 5.5–7.6 vs. 3.0–3.9, HR 0.79]	13.8 vs. 7.5 [70% CI 12.7–14.4 vs. 6.9–5.9]	6 vs. 10%	73 vs. 52% [70% CI 66–78 vs. 46–58]	
	Jackman et al. [70]	2	24	Bevacizumab + Erlotinib	Single arm	2.2 [95% CI 1.4–5.9]	5.8 [95% CI 2.8–10.1]	0%	50%	
	NCT02856425 (PEMBIB) [71]	1B	13	Pembrolizumab + Nintedanib	Single arm	-	16.3 [95% CI 4.3– not reached]	25%	58%	The 2 patients with mesothelioma had only PD
**Mesothelin**	Hassan R. et al. [77]	1	148	Anetumab ravtansine	Single arm	2.8 [95% CI 2.6–4.4]	-	8.9%	56.5	64 patients with mesothelioma Tumor response was evaluated in 138 patients.
	NCT02610140 [78]	2	248	Anetumab Ravtansine	Vinorelbine	4.3 vs. 4.5 [95% CI 4.1–5.2 vs. 4.1–5.8, HR 1.22]	9.5 vs. 11.6	8.4 vs. 6.1%	73.5 vs. 68.3%	
	Hassan et al. [80]	1	34	SS1P	Single arm			11.7%	67.6%	
	Hassan et al. [81]	2	11	SS1P/pentostatin/cyclophosphamide	Single arm			27.7%	54.5%	
	Haas et al. [82]	1	15	CART-meso cells	Single arm		2.1		73.3%	Pretreated patients
	Adusumilli et al. [83]	1	27	CAR-T +/− Cyclophosphamide +/− Pembrolizumab	Multi arm trial	17.7–23.9				Numbers are relative to CAR-T/cyclophosphamide arm and CAR/T cyclophosphamide/ pembrolizumab arm, respectively
	NCT01355965) [86]	1	18	CAR-T anti Mesothelon	Single arm					No efficacy results due to safety concerns
**Other targets**	Govindan et al. [87]	2	43	Gefitinib (anti EGFR)	Single arm	2.6 [95% CI 1.5–4.2]	6.8 [95% CI 3.5–10.]	4%	53%	
	Garland et al. [88]	2	63	Erlotinib (anti EGFR)	Single arm	2 [95% CI 2–4]	10 [95% CI 5–13]	0%	42% [95% CI 25–61%]	
	NCT02372227 [91]	1	21	VS-5584 (anti PI3K/mTOR) + VS-6063 (anti FAK)	Single arm	-	-	-	-	Data not published
	NCT01938443 [94]	1b	34	GSK2256098 (anti FAX) + Trametinib (anti MEK)	Single arm	2.6	-	0%	38%	
	NCT02860286 [95]	2	74	Tazemetostat (anti EZH2)	Single arm	4 [95% CI 2.8–4.1]	8 [95% CI 6.7–14]	3%	65%	
	MiST1 [96]	2a	26	Rucaparib (PARP inhibitor)	Arm 1 of multi arm MiST trial	4.1 [95% CI 2.8–5.7]	9.5 [95% CI 6.6– not reached]	4% at 24 weeks	23% at 24 weeks	
	MiST2 [98]	2	26	Abemaciclib (CDK4/6)	Arm 2 of multi arm MiST trial				54% at 12 weeks	

**Table 3 ijms-24-10415-t003:** Ongoing trials, completed trial with results not yet available or trials with unknown status, in all lines of therapy. Enrolled patients refers to the estimated enrolment.

	Trial	Phase/ Type of Trial	Enrolled Patients	Experimental Arm	Standard Arm/ Placebo/ Single Arm	Study Start Date	Estimated Completion Date	Setting
**Immunotherapy (anti PD-1/PD-L1)**	DREAM3R (NCT04334759) [13]	3	480	Pemetrexed/cis or carboplatin/durvalumab	Pemetrexed/ cis-carboplatin or Nivolumab/ ipilimumab	18 February 2020	December 2025	1st line
	IND227 (NCT02784171) [14]	2/3	520	Pemetrexed/cisplatin/ pembrolizumab or Pembrolizumab alone	Pemetrexed/ cisplatin	7 October 2016	June 2023	1st line
	ETOP BEAT-meso (NCT03762018) [15]	3	401	Pemetrexed/carboplatin/bevacizumab/ Atezolizumab	Pemetrexed/ carboplatin/ bevacizumab	30 April 2019	31 January 2024	1st line
**Vaccine**	NCT02649829 (MESODEC) [20]	1/2	28	WT1 Dendritic cells	Single arm	1 August 2017	December 2024	1st line
	NCT02395679 (MesoCancerVa) [22]	1	9	Dendritic Cells	Single arm	January 2015	December 2016	1st line
	NCT03610360 (DENIM) [25]	2/3	230	Dendritic Cells	Best supportive care	21 June 2018	15 February 2023	1st line
	NCT00299962 [59]	1	17	Ad.hIFN-β (BG00001)	Single arm	March 2006	October 2009	Pretreated patients
	NCT03710876 (INFINITE) [63]	3	53	rAd-IFN + Celecoxib + Gemcitabine	Celecoxib + Gemcitabine	21 January 2019	November 2024	Pretreated patients
	NCT04040231 [65]	1	10	Galinpepimut-S + Nivolumab	Single arm	24 July 2019	July 2023	Pretreated patients
	NCT04013334 [66]	2	12	MTG201 (Ad-SGE-REIC/Dkk-3)	Single arm	15 August 2019	1 January 2023	Pretreated patients
	NCT03546426 (MESOVAX) [67]	1b	18	Autologous dendritic cells + Pembrolizumab	Single arm	12 December 2019	January 2024	Pretreated patients
**Metabolism-based**	ATOMIC (NCT02709512) [29]	2/3	249	ADI-PEG 20/pemetrexed/cisplatin	Placebo/ pemetrexed/ cisplatin	1 August 2017	December 2022	1st line
	NCT05245500 [32]	1/2	339	MRTX1719	Single arm	2 June 2022	31 January 2024	Pretreated patients
**Immunotherapy (other targets than PD-L1/PD-1**	NCT02812875 [54]	1	71	CA-170	Single arm	May 2016	7 May 2020	Pretreated patients
	NCT03538028 [57]	1	22	INCAGN02385	Single arm	18 June 2018	7 October 2020	Pretreated patients
**Antiangiogenic**	NCT03502746 [69]	2	35	Ramucirumab + Nivolumab	Single arm	26 June 2018	June 2023	Pretreated patients
**Mesothelin**	NCT03126630 [79]	1/2	110	anetumab ravtansine/pembrolizumab	pembrolizumab	8 February 2018	16 March 2023	Pretreated patients
**Other targets**	NCT02758587 [90]	1/2A	59	Defactinib (anti FAK) + Pebrolizumab (anti PD-1)	Single arm	4 July 2017	December 2021	Pretreated patients
	NCT03917043 [93]	1	150	APG-2449 (anti FAK, ALK, ROS1)	Single arm	27 May 2019	February 2025	Pretreated patients
	NCT03654833 (MiST) [97]	2A	186	Bemcentinib (anti AXL) + Pembrolizumab	Arm 3 of multi arm MiST trial	28 January 2019	31 October 2023	Pretreated patients

## Data Availability

No new data were created or analyzed in this study. Data sharing is not applicable to this article.

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
