# Peer review of "Improvements in Systemic Therapies for Advanced Malignant Mesothelioma"

_ijms, 2023, doi:10.3390/ijms241310415_

Round 1

Reviewer 1 Report

Deiana et al. reviewed systemic therapies for advanced MPM. Although the subject was comprehensively reviewed, the manuscript needs to be edited so that it will be easy for readers to follow. For example, the paragraph with line 60 talked about CheckMate 743. It was not clear if the following two paragraphs also talked about CheckMate 743 or other studies. Authors should carefully review the content and edit the text throughout the manuscript. English editing is needed.

Just list several examples here:

1.       Some paragraphs only have one or two sentences. For example, paragraphs with lines 76, 79, 81, and 132. Some paragraphs can be combined since they were on the same topic or elaboration of the single sentence paragraph (e.g., paragraph with lines 132 and 134).  

2.       Line 223: “53” should be “53%”

3.       Line 329: “64” should be “64%”

4.       Line 295 sentence: needs to be rewritten

5.       Line 287: “New types of Immunotherapy” section listed 1, 2, and 3 trails. It was not clear why CA-170 was not listed, but others were listed. The structure or formatting was very confusing.

6.       Line 435: “Conclusions in second-line therapy” section was long. Some of the discussions can be moved to previous sections. 

FFormatting issue needs to be fixed. For example:

1.       Line 56

2.       Line 140 section: first paragraph

3.       Line 393 section

4.       Table 2

English editing is needed.  

Author Response

Response to reviewer 1

We would like to thank reviewer 1 for the comments/suggestions and for taking the time to review our paper.
Below is a list of the corrections we made:

  1. We altered the text extensively to make it easier to follow (major comment from reviewer 1).
  2. Editing and formatting issues were corrected.
  3. English editing was performed.
  4. Paragraphs with only few sentences have been edited and merged according to appropriate context.
  5. The section ‘conclusion on second line therapy’ has been shortened.
  6. Table 2 has been modified, both by fixing the formatting issues and by creating a third table on ongoing trials, making Table 2 dedicated exclusively on trials with published results.

Reviewer 2 Report

Dr. Chiara Deiana et al. summarized and reported on a clinical trial for the treatment of malignant pleural mesothelioma. In their review article, the results of the clinical trials are well summarized and may be helpful to the reader. However, there are a few corrections that need to be made.

My comments are listed below.

Major comments:

1.        The descriptions in Table 1 and Table 2 need to be modified to be more appropriate. For OS, PFS, and ORR, please describe the 95% confidence interval and hazard ratio (HR). Since DCR is considered less critical information than OS, PFS, and ORR, one option is to omit the description of DCR if there is more information in the Table. For "Ongoing clinical trials," it would be difficult to clearly understand if they are described together with the clinical trials for which results are available. Therefore, please consider creating a new Table 3, etc., and describe it separately from the clinical trials for which results are available. If you list the ongoing clinical trials in a separate table, please provide more details, such as when the trial started and when it is expected to end.

2.        Is the results of EAP not randomized controlled trials listed in Table 1 necessary? Is it possible that information such as survival time may not be collected as accurately as in other clinical trials? If so, please consider removing it from the Table.

Minor comments:

1.        Please correct "PD-1/D-L1" to "PD-1/PD-L1". (line 16)

2.        Please correct "Immunotherapy" to "Immunotherapy." (line 56)

3.        Please correct from italic to regular typeface. (line141 to 151)

4.        In "Table 2", the description after "Novel approaches" is strange; please correct it to match the description of "Immunotherapy" and "Chemotherapy."

Author Response

Response to reviewer 2

We would like to thank reviewer 2 for the comments/suggestions and for taking the time to review our paper.
Below is a list of the corrections we made:

  1. Major comment 1: when available, 95% confidence intervals and hazard ratios have been added to table 1 and 2. Additionally, a third table with ongoing trials/ trials with yet unpublished results has been created, with additional information on start and estimated completion date, as well as the setting (first line/pretreated patients).
  2. Major comment 2: the EAP has been removed from table 1.
  3. Minor comments 1-2-3 have all been accepted and corrected.
  4. Minor comments 4: table 2 had a formatting issue hiding the relative sections of the trial, it has now been edited for easier use.

Round 2

Reviewer 1 Report

Authors did a good job on revising the manuscript. It is acceptable for publication.

Reviewer 2 Report

The authors have carefully revised their paper to address my raised points. I believe the article is more understandable than the previous version of the document. I would like to extend my respect to the authors for their prompt and diligent response.